# Using Document Similarity Methods to create Parallel Datasets for Code Translation

## Abstract

Translating source code from one programming language to another is a critical, time-consuming task in modernizing legacy applications and codebases. Recent work in this space has drawn inspiration from the software naturalness hypothesis by applying natural language processing techniques towards automating the code translation task. However, due to the paucity of parallel data in this domain, supervised techniques have only been applied to a limited set of popular programming languages. To bypass this limitation, unsupervised neural machine translation techniques have been proposed to learn code translation using only monolingual corpora. In this work, we propose to use document similarity methods to create noisy parallel datasets of code, thus enabling supervised techniques to be applied for automated code translation without having to rely on the availability or expensive curation of parallel code datasets. We explore the noise tolerance of models trained on such automatically-created datasets and show that these models perform comparably to models trained on ground truth for reasonable levels of noise. Finally, we exhibit the practical utility of the proposed method by creating parallel datasets for languages beyond the ones explored in prior work, thus expanding the set of programming languages for automated code translation.

## 1 Introduction

As the pace of software development increases and the famous adage "software is eating the world" (Andreessen, 2011) is borne out, there is a corresponding increase in the amount of source code and number of software artefacts in active use for which support has lapsed. At the same time, the number of software professionals and programmers who can support and understand such code is unable to keep pace with the rate at which it is produced. This problem, while important when it comes to relatively modern programming languages (such as Java and Python), becomes even more pressing when it come to legacy languages (like COBOL) that mission-critical applications and systems are written in (Charette, 2020). In recent years, there have been multiple instances of organizations struggling to maintain their legacy systems and making considerable investments to upgrade them. In 2021 the Commonwealth Bank of Australia upgraded its core banking platform originally written in COBOL: this ultimately took 5 years and more than 1 Billion AUD to complete (Irrera, 2017). During the COVID-19 pandemic, software systems implemented in COBOL slowed down the release of US unemployment stimulus checks (Kelly, 2020), leaving governments scrambling to find COBOL experts who were already hard to come by. A recent study by the United States Government Accountability Office (Walsh, 2021) has identified 65 critical federal legacy systems in need of urgent modernization. Some of these systems are over 50 years old, and cost millions of dollars annually to operate and maintain.

Parallel to these developments are recent efforts at the intersection of software engineering, machine learning (ML), and natural language processing (NLP), which have posited the *naturalness hypothesis of software* (Hindle et al., 2016). The hypothesis states that *"...Software is a form of human communication; software corpora have similar statistical properties to natural language corpora; and these properties can be exploited to build better software engineering tools"* (Allamanis et al., 2018). This hypothesis has been used to extend breakthroughs and advances from various NLP sub-fields to software engineering tasks such as code translation. Prior works in the code translation domain have proposed the application of statistical, supervised, and unsupervised machine translation techniques to learn code translation models to varying degrees of success.

A key limitation of a majority of the proposed code translation approaches, however, is the lack of availability of parallel data for training. Unlike natural language, where a piece of text is verbatim translated in multiple languages – legal documents, parliamentary proceedings in multilingual societies – code is rarely implemented as is in multiple languages; thus making it hard to create parallel datasets. A few limited datasets – such as Java ↔ C# (Nguyen et al., 2013) and AVATAR for Java ↔ Python (Ahmad et al., 2021b) – are currently available. However, these are extremely limited in the number of programming language they cover, and manually curating a dataset for a specific use-case is impractical. To bypass this limitation, unsupervised techniques have been applied to the code translation task. Unsupervised techniques come with their own limitations however; and often, supervised techniques can outperform them when the source and target corpora are from different domains, the source and target languages use different scripts, and on low-resource language pairs, among other concerns (Kim et al., 2020; Marchisio et al., 2020).

It is for this reason that in this work, we focus on one of the main blockers impeding the application of supervised techniques to code translation: the availability of parallel corpora and datasets. Specifically, we propose to utilize document similarity methods to create parallel source code datasets that are noisy by design. In this work, we empirically demonstrate the effectiveness of document similarity methods in creating such parallel datasets with high levels of accuracy. Given that datasets created in this manner are bound to be noisy, we study the performance characteristics of models for code translation that have been trained on data with varying degrees of noise; and show that these models have considerable resistance to noise and perform well even with moderate amounts of noise. Finally, we demonstrate the practical utility of the proposed approach by training models to translate between 10 pairs of languages – a majority of which have not been looked at in prior work.

## 2 RELATED WORK

**Code translation datasets:** Typical methods for creating parallel datasets for code translation have either relied on the availability of open-sourced projects with implementations in multiple languages, or on the existence of transpilers. The earliest widely-used large-scale dataset for code translation was for Java ↔ C# (Nguyen et al., 2013) translation, created by indexing open-sourced projects implemented in both languages. Aggarwal et al. (2015) used the Python 2to3 [1] transpiler to create a dataset; while Chen et al. (2018) used CoffeeScript's compiler (which compiles down to JavaScript) to create a parallel dataset. More recently, Ahmad et al. (2021b) released AVATAR – a parallel corpus of Java to Python manually curated through submissions on competitive programming websites. Publicly available datasets for code translation are however extremely limited, and manually curating these datasets for a specific use-case is expensive and often impractical.

**Source-to-Source translation:** The earliest code translation models were rule-based systems, operating on handcrafted rules. These systems require a lot of effort to build, are not easily extendable to other languages, and are also outperformed by neural techniques. Some of these systems are: `Java2CSharp` [2], `Java2Python` [3], SmallTalk to C (Yasumatsu & Doi, 1995), Cobol to Java (Mossienko, 2003), and Tangible Software Solutions [4] (VB.NET, C#, Java, C++, and Python). Moving away from rule-based systems, Nguyen et al. (2013), Karaivanov et al. (2014), and Nguyen et al. (2014) applied different versions of Phrase-Based Statistical Machine Translation to translate between Java and C#. Chen et al. (2018) proposed a tree-to-tree neural network to translate the parsed tree of the source code into the target code parse tree. The aforementioned supervised techniques have all been benchmarked on the Java ↔ C# dataset, and are limited by the availability of parallel datasets. To bypass this limitation, Roziere et al. (2020) used unsupervised neural machine translation techniques to translate between languages using only monolingual corpora, and showed impressive results for translation between Java, C++, and Python. While Roziere et al. (2020) trained the model specifically for code translation, large language models – such as GPT-2 (Radford et al., 2019), GPT-3 (Brown et al., 2020), and Codex (Chen et al., 2021) – have also been shown to have some competence in generating code (Hendrycks et al., 2021).

**Parallel corpus mining:** Prior work in natural language research has looked at various ways of creating parallel corpora from a non-parallel corpus. Munteanu & Marcu (2005) train a maximum

---

[1] https://docs.python.org/3/library/2to3.html
[2] https://sourceforge.net/projects/j2cstranslator/
[3] https://github.com/natural/java2python
[4] https://www.tangiblesoftwaresolutions.com/

entropy classifier to identify if two given sentences are translations of each other. They extract parallel data from large-scale Chinese, Arabic, and English non-parallel newspaper corpora, and show improvement in model performance when trained with a combination of a small parallel corpus and the extracted dataset. Uszkoreit et al. (2010) describe a system that uses n-gram features to mine parallel documents from a billion-scale corpus. Smith et al. (2010) focus on aligning Wikipedia documents by creating features suitable for such documents. Artetxe & Schwenk (2019) utilize specific scoring functions based on multilingual sentence embeddings to create parallel corpora, and Hangya & Fraser (2019) rely on continuous parallel segments rather than word similarities to find parallel sentences. Banón et al. (2020) released the largest publicly available parallel corpora of sentences (223 million parallel sentences) by aligning sentences from data crawled over the web. There is a substantial precedence of parallel corpus mining in the natural language domain; however, such studies in the code translation domain are non-existent.

**Machine Translation using noisy data:** Prior studies have aimed to study the impact of noise on the performance of machine translation systems. Formiga & Fonollosa (2012) study the impact of misspelled words on the performance of Statistical Machine Translation and suggest strategies to deal with them, while Goutte et al. (2012) study the impact of sentence alignment errors on the performance of SMT. Further, Khayrallah & Koehn (2018) define 5 categories of artificial noise in Neural Machine Translation, and study the impact each of these types has on performance. We motivate our work from these prior efforts in order to study the impact that noise has on the performance of code translation models.

## 3 Proposed Method

In this work, we propose to utilize document similarity methods to create noisy parallel datasets for code translation. We refer to the datasets created in this manner as "noisy" because unlike manually curated datasets, there is no guarantee of a parallel implementation of the specific source file/code being available in the corpus: this may result in near-similar code samples being paired as ground truth examples instead. Algorithm 1 presents the proposed approach as pseudocode.

The algorithm expects two non-parallel sets of documents $D = \{d_1, \cdots, d_n\}$ and $D^{'} = \{d^{'}_1, \cdots, d^{'}_m\}$ as input. Within the context of our work, the documents in these two sets represent code samples from two distinct programming languages. Along with the documents, the algorithm also expects a similarity measure $M(d, d^{'})$ as input, to compare two given documents for similarity. A lower score from the similarity measure indicates higher similarity between documents. Finally, the algorithm expects a similarity threshold $\delta$ to help keep only sufficiently similar documents in the resulting parallel corpus. Thereafter, the algorithm follows a simple procedure of

---

**Algorithm 1** Creating parallel code corpus

1: **CreateParallelCorpora**($D, D^{'}, M, \delta$)
2: initialize $P = \{\}$
3: **for** $i = 1$ to $|D|$ **do**
4:     $D_{sim}$ = GetSimilarDocuments($d_i, D^{'}, M$)
5:     **for** $(d_i, d^{'}_j)$ in $D_{sim}$ **do**
6:         **if** $(\cdot, d^{'}_j) \notin P$ **and** $M(d_i, d^{'}_j) \leq \delta$ **then**
7:             $P = P \cup (d_i, d^{'}_j)$
8:             break
9: $D_{res} = sort((d_1, d_2) \in P, key = M(d_1, d_2))$
10: **Return:** $D_{res}$

---

iterating over all documents; finding the most similar documents in the target set; and adding the newly found similar document pairs to the result only if the target document has not been paired before, and if the similarity is below the threshold value. Once all the documents are iterated upon, the algorithm produces a list of unique pairs of code segments (documents) ordered by their similarity, ready to be used for downstream tasks.

## 4 Experimental setup

To better understand the effectiveness and practical utility of the method proposed in Section 3, we devise 3 research questions and design experiments to empirically answer them (see Section 5). In this section, we briefly summarize the different document similarity methods, datasets, pre-trained models, and evaluation metrics we use in our experiments.

## 4.1 DOCUMENT SIMILARITY METHODS

**TF-IDF:**  TF-IDF (Salton & Buckley, 1988) computes the product of the term frequency (TF) (fraction of times a term appears in a document) with the inverse document frequency (IDF) (logarithm of the inverse fraction of documents a particular token occurs in). The cosine similarity between the document vectors thus created computes the similarity of documents.

**Okapi-BM25:**  The Okapi-BM25 model (Robertson et al., 1995) uses the following scoring function (Equation 1) to score the importance of a word $w$ in a document $D$. Here, $IDF(w)$ represents the inverse document frequency of the word $w$, $TF(w, D)$ represents the term frequency of the word $w$ in the document $D$, $|D|$ and $D_{avg}$ are the lengths of the current document and the average document lengths respectively, and $k_1$ and $b$ are free parameters of the model.

$$BM25(w, D) = IDF(w) \times \frac{TF(w, D)(k_1 + 1)}{TF(w, D) + k_1(1 - b + b\frac{|D|}{D_{avg}})} \tag{1}$$

**Latent Dirichlet Allocation (LDA):**  LDA (Blei et al., 2003) is a hierarchical generative Bayesian model that models each document as a finite mixture over an underlying set of topics. The cosine similarity of the topic distribution of two documents computes their similarity.

**Latent Semantic Indexing (LSI):**  LSI (Deerwester et al., 1990) computes the Singular Value Decomposition of the Bag of Words representation of documents. The cosine similarity of the decomposed vectors of documents computes their similarity.

**Word Movers Distance (WMD):**  WMD (Kusner et al., 2015) models the document distance problem as a variant of the Earth Movers Distance (Monge, 1781; Rubner et al., 1998) and solves the optimization problem defined in Equation 2.

$$\min_{T \geq 0} \sum_{i,j=1}^{n} T_{ij}c(i, j)$$
$$\text{subject to: } \sum_{j=1}^{n} T_{ij} = d_i \quad \forall i \in \{1, \cdots, n\} \tag{2}$$
$$\sum_{i=1}^{n} T_{ij} = d_j \quad \forall j \in \{1, \cdots, n\}$$

Here, $T \in \mathbb{R}^{n \times n}$ is a flow matrix where $T_{ij} \geq 0$ and denotes how much of word $i$ in document $d$ travels to word $j$ in document $d^{'}$. $c(i, j) = \|x_i - x_j\|_2$ is the cost associated with travelling from one word to another, $d$ and $d^{'}$ are the nBOW representations of the documents, and $X \in \mathbb{R}^{d \times n}$ is the word embedding matrix where $x_i \in \mathbb{R}^d$ represents the $d$-dimensional embedding of the $i^{th}$ word.

## 4.2 DATASETS

For the experiments whose results are detailed in Section 5, we utilize the following datasets. We provide representative code samples and statistics from the datasets in Appendix A.

**Java $\leftrightarrow$ C#:**  The Java $\leftrightarrow$ C# dataset is one of the earliest large-scale datasets introduced for the code translation task (Nguyen et al., 2013; Zhong et al., 2010). It is created by indexing several open-source projects which have both Java and C# implementations, and pairing methods with the same file name and method name. The earlier version of this data was created by indexing the `db4o` and `Lucene` projects. More recently however, Chen et al. (2018) indexed 6 open-sourced projects to create the dataset. We use the version provided by Chen et al. (2018) in our work.

**Java $\leftrightarrow$ Python, Java $\leftrightarrow$ C++, and Python $\leftrightarrow$ C++:**  Roziere et al. (2020) extracted parallel functions in C++, Python, and Java from the online competitive programming platform GeeksForGeeks[5],

---

[5]https://practice.geeksforgeeks.org/

and used these code samples as validation and test sets. We, however, concatenate the two datasets and use the unified dataset for our experiments. The code samples in this dataset are function-scope code samples that solve an algorithmic problem.

**CodeNet:** Project CodeNet (Puri et al., 2021) is a recently released large-scale AI for Code dataset, created by indexing two online competitive programming websites. The dataset is organized into about 4000 different problem sets, and contains a little under 14 million total solutions in 55 programming languages. Besides providing the code samples, CodeNet also provides input-output pairs to evaluate solutions to the problem sets.

## 4.3 MODELS

**CodeBERT:** CodeBERT (Feng et al., 2020) is a Transformer (Vaswani et al., 2017) based model, pre-trained on a unimodal data of function-level code samples, and a bimodal data of code and the associated documentation in natural language. The pre-training data contains code samples in Go, Java, JavaScript, PHP, Python, and Ruby, and is trained using the Masked Language Modeling (Devlin et al., 2019) (MLM) and the Replaced Token Detection (Clark et al., 2019) objectives.

**GraphCodeBERT:** GraphCodeBERT (Guo et al., 2020) is a Transformer based model for code that also considers the inherent structure in code by integrating the data flow in the pre-training stage. The model is trained on the CodeSearchNet dataset (Husain et al., 2019) using the MLM, Edge Prediction, and Node Alignment objectives.

**PLBART:** PLBART (Ahmad et al., 2021a) is a BART (Lewis et al., 2020) based model pre-trained on over 700 million Java, Python, and natural language documents collected from open-sourced code on Github and posts on StackOverflow. The model is pre-trained via denoising autoencoding, where the model learns to reconstruct input corrupted by a noise function. The authors use three noising strategies: token masking, token deletion, and token infilling to create the corrupted inputs.

## 4.4 EVALUATION METRICS

**BLEU score:** BLEU score (Papineni et al., 2002) is a common automatic evaluation metric for machine-generated text, and exhibits a high correlation with human judgment of quality. BLEU score is computed as the overlapping fraction of n-grams between the machine-generated text and the reference text. The metric has however been shown to not be a reliable measure for source code (Ren et al., 2020; Allamanis et al., 2018; Austin et al., 2021).

**CodeBLEU score:** Ren et al. (2020) propose the CodeBLEU score to leverage the tree structure and semantic information in code. It is computed as a combination of the standard BLEU score, weighted n-gram match, syntactic abstract syntax tree match, and the semantic data flow match.

**Exact Match (EM):** EM (Nguyen et al., 2013) evaluates if the generated code matches exactly to the reference code.

**Computational Accuracy @ $k$ (CA@$k$):** Recent work in code synthesis has adopted the CA@$k$ metric (Austin et al., 2021; Roziere et al., 2020) to evaluate code generation models. To compute CA@$k$, $k$ samples are generated from the model, and the problem is considered solved if any of the generated $k$ samples pass the unit tests associated with the problem.

## 5 RESEARCH QUESTIONS AND RESULTS

To validate the central hypothesis of this paper – using document similarity methods to create datasets for supervised training of code translation models – we define and seek answers to the following research questions (RQ):

RQ1:  How accurate are document similarity methods in creating parallel datasets for code?

RQ2:  Given the created dataset will be noisy, what is the effect of varying degrees of noise on code translation models?

RQ3:  Can the proposed method be used in practice to train models for programming languages not explored in prior work?

## 5.1 **RQ1**: EFFICACY OF DOCUMENT SIMILARITY METHODS

We start our analysis by examining how effective document similarity methods are in creating code translation datasets. For this experiment, we utilize 4 datasets with known ground-truth mapping between pairs of programming languages – Java ↔ C#, Java ↔ Python, Java ↔ C++, and Python ↔ C++. For each of these datasets, we create a parallel dataset using 5 different document similarity methods, and compute the match accuracy as the number of correctly matched code samples.

We summarize the results for this experiment in Table 1, and observe that similarity methods that operate in a latent space (such as LDA and LSI) perform much worse than methods that operate in the original space (such as TF-IDF, Okapi-BM25, and WMD). We posit that because code is written in a more formal language than natural language, and each data sample in the datasets used in this experiment implements an independent unique function, there is likely no underlying topic or latent semantic associations that can be captured by LSI and LDA. Therefore these methods perform worse than methods that directly utilize the tokens in the original space.

Table 1: Match accuracy of parallel datasets created using various document similarity methods. Match accuracy computes True if the matched code sample is the same as the code sample in the ground truth dataset.

|  | **Java ↔ C#** | **Java ↔ Python** | **Java ↔ C++** | **Python ↔ C++** |
|---|---|---|---|---|
| **N ($\rightarrow$)** | 10,300 | 1,418 | 1,418 | 1,418 |
| LDA | 47.21% | 21.44% | 35.83% | 16.64% |
| LSI | 57.21% | 66.08% | 87.66% | 78.84% |
| TF-IDF | 87.36% | 86.10% | 94.08% | 89.35% |
| Okapi-BM25 | 87.86% | 89.91% | **95.77%** | 89.99% |
| WMD | **89.53%** | **91.04%** | 95.06% | **94.08%** |

We note that the datasets used for the experiments in Table 1 are not true representatives of code we expect to find while using this method in practice. This is due to the fact that the 4 datasets used contain code samples for which a true parallel implementation in the other language exists. When trying to create a parallel dataset from code collected in the wild, we cannot be sure of the availability of a true parallel implementation, which might affect the performance of the proposed method. Thus, to account for this phenomenon, we conduct a similar experiment with the CodeNet dataset. Since code samples in the CodeNet dataset are not parallel implementations, this gives us a better idea of the effectiveness of document similarity methods in creating parallel datasets from code in the wild. We select 6 languages and randomly sub-sample 50 problems from the dataset. For each of the code sample in each of the 50 sampled problem sets, we create a parallel dataset using 3 different document similarity methods. Since we do not have a ground-truth parallel implementation available for the CodeNet dataset, we cannot compute the match accuracy like we did in the previous experiment. We therefore compute the pseudo-match accuracy instead. The pseudo-match accuracy computes True if the matched code sample is a solution of the same problem set, and False otherwise.

We report the pseudo-match accuracy results for this experiment in Table 2. We note that while the TF-IDF and Okapi-BM25 methods performed well in the previous experiment, their performance varies greatly for this experiment. Datasets created using TF-IDF and Okapi-BM25 methods are matched to the correct problem set with as little as 30% accuracy and as high as 70% accuracy in some cases. Datasets created using the WMD method however achieve a high match accuracy for both experiments (Table 1 and Table 2).

From the experiments designed to answer RQ1, we conclude that document similarity methods are capable of creating parallel datasets of code with a significantly high degree of match accuracy. Specifically, the WMD metric seems to be quite adept at delineating datasets for code translation.

## 5.2 **RQ2**: NOISE TOLERANCE OF MODELS TRAINED ON CODE

In the previous section, we showed that document similarity methods – and specifically the Word Movers Distance (WMD) – are quite adept at creating parallel datasets for code. However, datasets created in this manner contain errors; therefore in this section, we seek to understand the effect that varying the degree of such noise has on the performance of code translation models.

Table 2: Pseudo-match accuracy of datasets created by different document similarity methods on 50 subsampled problems from CodeNet. Pseudo-match accuracy computes True if the matched code sample is from the same problem set, and False otherwise.

| | Source language: Go | | | | | | Source language: Java | | | | |
|---|---|---|---|---|---|---|---|---|---|---|---|
| | Java | JavaScript | PHP | Python | Ruby | | Go | JavaScript | PHP | Python | Ruby |
| TF-IDF | 29.56% | 41.91% | 30.70% | 40.35% | 27.07% | TF-IDF | 65.05% | 53.40% | 29.41% | 31.32% | 23.47% |
| BM25 | 50.93% | 49.07% | 50.52% | 36.93% | 35.37% | BM25 | 63.94% | 73.46% | 34.11% | 51.63% | 36.09% |
| WMD | **71.47%** | **55.70%** | **51.35%** | **60.89%** | **45.12%** | WMD | **79.30%** | **73.93%** | **57.51%** | **66.49%** | **56.06%** |

| | Source language: JavaScript | | | | | | Source language: PHP | | | | |
|---|---|---|---|---|---|---|---|---|---|---|---|
| | Go | Java | PHP | Python | Ruby | | Go | Java | JavaScript | Python | Ruby |
| TF-IDF | 50.93% | 46.64% | 44.89% | 41.18% | 55.34% | TF-IDF | 64.31% | 27.49% | 45.66% | 26.04% | 37.62% |
| BM25 | 51.74% | 58.12% | 56.26% | 59.74% | 61.60% | BM25 | 62.54% | 50.48% | 48.71% | 27.97% | 55.63% |
| WMD | **66.70%** | **74.71%** | **67.52%** | **70.88%** | **73.55%** | WMD | **71.38%** | **61.25%** | **73.79%** | **85.05%** | **84.08%** |

| | Source language: Python | | | | | | Source language: Ruby | | | | |
|---|---|---|---|---|---|---|---|---|---|---|---|
| | Go | Java | JavaScript | PHP | Ruby | | Go | Java | JavaScript | PHP | Python |
| TF-IDF | 67.19% | 52.79% | 66.67% | 69.54% | 72.46% | TF-IDF | 61.94% | 51.53% | 69.85% | 61.42% | 67.51% |
| BM25 | 73.52% | 57.53% | 74.13% | 73.33% | 80.28% | BM25 | 67.33% | **58.66%** | 74.37% | 62.22% | 82.45% |
| WMD | **82.08%** | **73.75%** | **77.39%** | **79.59%** | **87.31%** | WMD | **68.80%** | 54.35% | **78.59%** | **77.05%** | **90.39%** |

We use the CodeBERT and the GraphCodeBERT pre-trained models for this experiments, and fine-tune these models on different pairings of the Java ↔ C# dataset (Nguyen et al., 2013) created using the different document similarity methods. We compare the performance of models trained on these paired datasets with models trained on the ground-truth dataset, and a random baseline with random pairings of code samples from the two programming languages. Following Ahmad et al. (2021a), we compute the BLEU score, CodeBLEU score, and the Exact Match score.

The results for this experiment are summarized in Table 3. We additionally refer the reader to Table 1 to see the corresponding match accuracy of the different document similarity methods. Interestingly, we find that models trained on noisy code datasets have a certain degree of resistance to noise; and while the performance drops with increasing levels of noise, the degradation is not sudden. Even with high levels of noise, the models perform considerably well. With a high-performing method such as the Word Movers Distance (WMD) – with about 90% match accuracy – the degradation in performance is roughly 1 percentage point across the three measures and for both directions of translation. For methods with a higher level of noise – such as LDA with 47.21% match accuracy – while the performance goes down significantly, it is still significantly higher than the performance of the random baseline. We posit that although noisy datasets create pairs of code with incorrect parallel implementations, much of the semantics is still retained by the dataset due to the formal nature of programming languages. For example, even if code samples are incorrectly paired, the syntax for function and variable definition, code blocks, and indentation stays the same and is preserved. This allows the model to learn the translation task to a certain degree.

Table 3: Model performance on the Java ↔ C# dataset matched using various document similarity methods. Each method introduces a different amount of noise in the resulting dataset (see Table 1) thereby affecting the performance.

| | | Java ⟶ C# | | | C# ⟶ Java | | |
|---|---|---|---|---|---|---|---|
| | | BLEU | CodeBLEU | EM | BLEU | CodeBLEU | EM |
| | Random baseline | 12.2 | 31.71 | 0.0% | 4.4 | 16.56 | 0.0% |
| | LDA | 57.55 | 65.97 | 33.2% | 43.75 | 55.39 | 23.7% |
| | LSI | 71.8 | 77.64 | 47.9% | 52.44 | 66.26 | 30.9% |
| **CodeBERT** | Okapi-BM25 | 79.39 | 83.54 | 59.5% | 73.83 | 80.64 | 57.6% |
| | TF-IDF | 78.78 | 82.70 | 58.1% | 72.52 | 77.34 | 57.1% |
| | WMD | 79.59 | 83.85 | 58.2% | 75.16 | 80.93 | 59.2% |
| | Ground-truth | 80.83 | 84.86 | 60.6% | 75.84 | 81.64 | 59.9% |
| | Random baseline | 6.88 | 20.09 | 0.0% | 2.94 | 14.31 | 0.0% |
| | LDA | 60.34 | 67.91 | 37.3% | 47.14 | 58.79 | 26.3% |
| | LSI | 73.74 | 79.30 | 49.1% | 52.86 | 66.49 | 30.5% |
| **GraphCodeBERT** | TF-IDF | 79.14 | 83.04 | 57.8% | 73.14 | 77.55 | 57.9% |
| | Okapi-BM25 | 79.65 | 83.42 | 59.4% | 74.24 | 80.56 | 58.0% |
| | WMD | 79.47 | 83.72 | 59.0 % | 75.63 | 81.06 | 60.2 % |
| | Ground-truth | 80.89 | 85.05 | 61.1 % | 76.76 | 82.03 | 62.3 % |

While the preceding experiment allowed us to understand the performance of models trained on noisy datasets created using different document similarity methods, we wish to understand the per-

formance characteristics of models trained with varying levels of noise on a more granular level. Thus we create datasets for translation between Java and C# by artificially injecting noise of varying levels. For explanation, in a dataset with $x\%$ noise level, we randomly misalign $x\%$ of code samples in that dataset, while keeping the remaining code samples correctly paired. We then train the GraphCodeBERT model on these datasets, and compute the BLEU score, CodeBLEU score, and the Exact Match (EM) score. Figure 1 shows the performance curve with levels of noise varying from 0% (ground-truth dataset) to 100% (complete random pairings). We observe that the degradation in the performance is gradual for initial levels of noise – compared to performance at 0% noise, the performance at about 30% noise goes down slowly by about 20 percentage points across all measures and both ways of translation. Post this 30% noise level, we see a sharper degradation in performance; and post 70% noise, we observe that the performance is just slightly better than the performance at 100% noise.

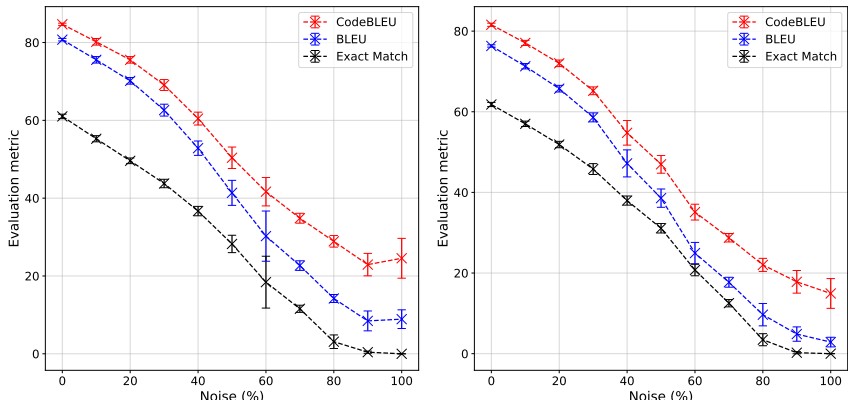

Figure 1: Noise performance curve for GraphCodeBERT model fine-tuned for Java $\rightarrow$ C# translation (left) and C# $\rightarrow$ Java translation (right). While the performance decreases with increasing levels of noise, the degradation is gradual for initial levels of noise.

Through the experiments in Table 3 and in Figure 1, we get a better idea of the performance characteristics of models for code under varying levels of noise. We conclude that the performance of these models is not severely affected with moderate amounts of noise; therefore when creating datasets from code in the wild where we expect a certain amount of noise, we can expect the models to perform reasonably well.

### 5.3  RQ3: Translating between a wider set of programming languages

In Section 5.1, we concluded that document similarity methods are adept at creating parallel datasets for code with acceptable levels of noise; and in Section 5.2, we concluded that models trained on noisy datasets of code perform reasonably well under moderate levels of noise. In this section, we take advantage of these two findings and demonstrate the practical utility of the proposed method by creating noisy code translation datasets for languages not explored previously in the literature; and by training models for translating between these languages.

For this experiment, we utilize the CodeNet dataset by creating noisy parallel datasets between the following 10 languages – C, C#, C++, Go, Java, JavaScript, PHP, Python, Ruby, and Scala. We choose these languages in order of their frequency in the CodeNet dataset, thereby maximizing the number of data samples we can potentially create. We also sub-sample about 2500 problem sets from the original 4000 problem sets from the CodeNet dataset for computational reasons. Thereafter, we match the solutions in one programming language to another for each of the 2500 sub-sampled problem sets using the WMD metric. Since the PLBART model can only translate sequences with a maximum of 512 tokens, we only match code samples with less than 512 tokens. We additionally use a similarity threshold of 3.0 and filter samples accordingly. In Appendix B.3, we provide statistics of the final dataset along with some representative code samples and their corresponding similarity scores. To create the test set for the language pairs, we sub-sample 100 problems that are not seen in the training and the validation set, and randomly sample 5 different implementations in the source language from each of the 100 problem sets for a final test set size of 500 code samples.

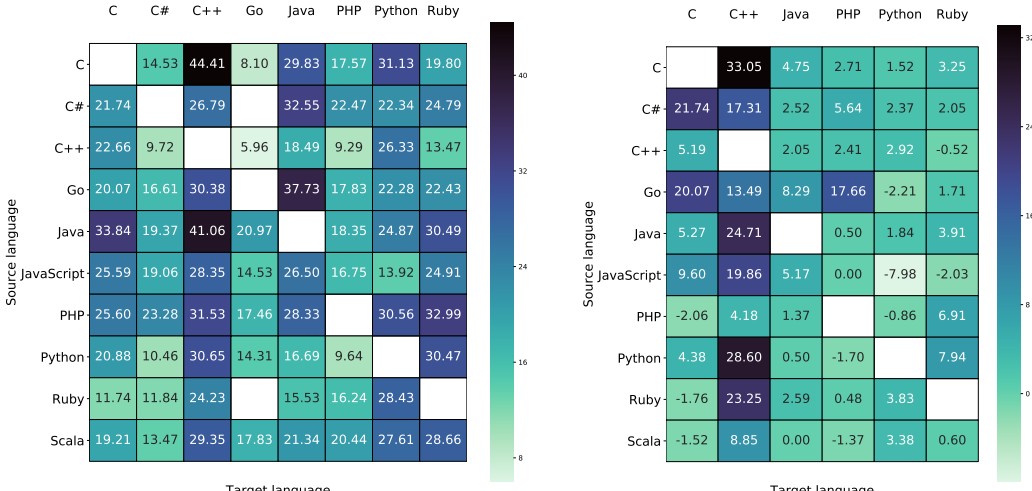

Figure 2: CA@5 for PLBART model fine-tuned on code translation dataset created from the CodeNet dataset using the WMD metric.

Figure 3: Difference between the CA@5 of PLBART trained using data created using the proposed method and the CA@5 of PLBART trained using randomly matched dataset.

We fine-tune the PLBART model on the matched training data for each language pair, and evaluate the computational accuracy @ 5 on the test set. We compare the performance of this fine-tuned model against a model fine-tuned on a dataset created by randomly matching solutions from each problem set rather than using the WMD metric. To compare these two models fairly, we keep the dataset sizes, problem sets, and all the hyperparameters constant across the two training procedures. The CA@5 results for the model trained on the WMD-matched dataset are shown in Figure 2; while Figure 3 shows the difference in the performance of the model fine-tuned using the WMD-matched data and the performance of the model fine-tuned using the randomly matched data. We present some of the model generated code samples in Appendix B.4.

Overall, we observe that models trained using the WMD-matched datasets achieve noteworthy performance across language pairs. More importantly, when compared with models trained on randomly paired data, we see substantial improvements for a majority of the language pairs. While the common language pairs, such as C → C++, Python → C++, Ruby → C++ see the biggest improvements, more obscure language pairs such as PHP → C++, PHP → Ruby, JavaScript → C++, and Python → Ruby also demonstrate substantial improvements over their random counterparts. This leads us to the conclusion that the proposed method is a viable way of creating high-quality datasets for code translation, thereby alleviating the paucity of training data in the domain.

## 6 DISCUSSION & CONCLUSION

Modernizing legacy applications into a new programming language is a process that requires a lot of time, intellect, and monetary investment. Automatic code translation techniques have the potential to speed up this process, and to reduce the human effort required by either working in tandem with humans or automatically translating legacy code to a modern language of choice. While multiple techniques have been proposed to improve the quality of code translation, their practical utility is hampered due to the limited availability of parallel data required to train these models between languages of choice. In this work, we proposed a simple technique to utilize document similarity methods to create noisy datasets for code translation; and demonstrated that models for code have a certain amount of tolerance for noise and perform well even under significant amounts of noise. We specifically demonstrated the effectiveness of the Word Movers Distance (WMD) metric in creating parallel datasets between numerous language pairs that have not been explored in prior literature; and showed significantly improved model performances as compared to models trained on randomly matched datasets. Future work will explore better metrics in terms of both match accuracy and computational efficiency, thereby further reducing the noise in the dataset; and incorporating the similarity score in the model to weight samples according to their computed similarity.

# 7 ETHICS STATEMENT

One of the major ethical points to consider when dealing with the automatic creation and translation of source code centers around the effects on humans: both humans who create and maintain code for a living; and humans that are affected by the decisions and outcomes produced by the execution of such code. For the former concern, our work merely seeks to align pre-existing bits of open-sourced code so that downstream data-hungry techniques may have more reasonable approximations of correct and on-purpose code to learn from. Our work does not replace jobs that humans are trained to do and more adept at; and indeed defers to and takes inspiration from prior studies (Weisz et al., 2021) that show that human generators of code are very likely to engage in partnerships with automatically learned models to produce or maintain code better. For the latter concern, we acknowledge that it is possible to use the output of artefacts from our work in downstream systems that can produce automatic code with little to no oversight. Similar to work on examining the effects of large language models for human natural languages (Bender et al., 2021), much attention is needed where it comes to automatic code generation and translation techniques and models. We look forward to studying some of these issues in partnership with colleagues in the future.

# 8 REPRODUCIBILITY STATEMENT

To allow for the reproducibility of experiments conducted in this work, we provide the source code of the experiments in the supplementary material attached with the submission. The supplementary material is grouped by the three research questions we define in our work. Each set of files within a folder corresponding to a research question contains the source code to the respective experiments. Another critical aspect of our work is the creation of parallel code translation datasets across many languages from the CodeNet dataset. Along with the source code, we also provide the train, validation, and test data sets for a small subset of language pairs in the attached supplementary material. We provide a small subset due to the limitations on the amount of data that can be provided as supplementary material. However, we plan to release the complete noisy dataset we created for our experiments with the final version of the paper.

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

## A REPRESENTATIVE CODE SAMPLES FROM UTILIZED DATASETS

In this section, we provide representative code samples from the datasets we use in this work. Listings 1 and 2 are data samples from the Java ↔ C# dataset. Listings 3 and 4 are data samples from the Java ↔ Python dataset. Listings 5 and 6 are data samples from the Java ↔ C++ dataset. Listings 7 and 8 are data samples from the C++ ↔ Python dataset. Finally, listings 9, 10, 11, 12, 13, 14, 15, 16 provide code samples from one particular problem set from the CodeNet dataset.

Listing 1: Java ↔ C#: Java code sample

```java
public static Cell getCell(Row row, int columnIndex)
{
    Cell cell = row.getCell(columnIndex);
    if (cell == null)
    {
        cell = row.createCell(columnIndex);
    }
    return cell;
}
```

Listing 2: Java ↔ C#: C# code sample

```csharp
public static ICell GetCell(IRow row, int column)
{
    ICell cell = row.GetCell(column);
    if (cell == null)
    {
        cell = row.CreateCell(column);
    }
    return cell;
}
```

Listing 3: Java ↔ Python: Java code sample

```java
static int binaryToDecimal ( int n )
{
    int num = n ;
    int dec_value = 0 ;
    int base = 1 ;
    int temp = num ;
    while ( temp > 0 ) {
        int last_digit = temp % 10 ;
        temp = temp / 10 ;
        dec_value += last_digit * base ;
        base = base * 2 ;
    }
    return dec_value ;
}
```

Listing 4: Java ↔ Python: Python code sample

```python
def binaryToDecimal ( n ) :
    num = n ;
    dec_value = 0 ;
    base = 1 ;
    temp = num ;
    while ( temp ) :
        last_digit = temp % 10 ;
        temp = int ( temp / 10 ) ;
        dec_value += last_digit * base ;
        base = base * 2 ;
    return dec_value ;
```

Listing 5: Java ↔ C++: Java code sample

```java
static int findS ( int s ) {
    int sum = 0 ;
    for ( int n = 1 ;
    sum < s ;
    n ++ ) {
        sum += n * n ;
        if ( sum == s ) return n ;
    }
    return - 1 ;
}
```

Listing 6: Java ↔ C++: C++ code sample

```cpp
int findS ( int s ) {
    int sum = 0;
    for ( int n = 1;
    sum < s;
    n ++ ) {
        sum += n * n;
        if ( sum == s ) return n;
    }
    return - 1;
}
```

Listing 7: C++ ↔ Python: C++ code sample

```cpp
void printDistinct ( int arr [ ], int n ) {
    sort ( arr, arr + n );
    for ( int i = 0;
    i < n;
    i ++ ) {
        while ( i < n - 1 && arr [ i ] == arr [ i +
            ↪ 1 ] ) i ++;
        cout << arr [ i ] << " ";
    }
}
```

Listing 8: C++ ↔ Python: Python code sample

```python
def printDistinct ( arr , n ) :
    arr.sort ( ) ;
    for i in range ( n ) :
        if ( i < n - 1 and arr [ i ] == arr [ i + 1
            ↪ ] ) :
            while ( i < n - 1 and ( arr [ i ] == arr
                ↪ [ i + 1 ] ) ) :
                i += 1 ;
        else :
            print ( arr [ i ] , end = " " ) ;
```

Listing 9: CodeNet: C code sample

```c
#include<stdio.h>
int main(){
  int i,j;
  for(i=1;i<10;i++){
    for(j=1;j<10;j++){
      printf("%dx%d=%d\n",i,j,i*j);
    }
  }
  return 0;
}
```

Listing 10: CodeNet: C# code sample

```csharp
using System;
class test
{
    static void Main()
    {for (int i = 1; i < 10; i++) for (int j = 1; j
        ↪ < 10; j++) Console.WriteLine(i+"x"+j+"="
        ↪ +i*j);}
}
```

Listing 11: CodeNet: Go code sample

```go
package main

import "fmt"

func main() {
        for i := 1; i < 10; i++ {
                for j := 1; j < 10; j++ {
                        fmt.Printf("%dx%d=%d\n", i,
                        ↪ j, i*j)
                }
        }
}
```

Listing 12: CodeNet: Java code sample

```java
public class Main
{
  public static void main(String[] args)
  {
    for(int a=1;a<=9;a++){
      for(int b=1;b<=9;b++){
        System.out.println(a+"x"+b+"="+a*b);
      }
    }
  }
}
```

Listing 13: CodeNet: JavaScript code sample

```javascript
for (var i=1; i<10; i++) {
   for (var j=1; j<10; j++) {
      console.log(i + 'x' + j + '=' + i*j)
   }
}
```

Listing 14: CodeNet: PHP code sample

```php
<?php
for($i=1;$i<10;$i++){
for($j=1;$j<10;$j++){
echo $i."x".$j."=".$i*$j."\n";
}
}
```

Listing 15: CodeNet: Ruby code sample

```ruby
# Your code here!

9.times{|i|
  i=i+1
  9.times{|j|
  j=j+1
  puts i.to_s+'x'+j.to_s+'='+(i*j).to_s
}}
```

Listing 16: CodeNet: Scala code sample

```scala
object Main{
  def main(args: Array[String]){
    for(i <- 1 to 9){
      for(j <- 1 to 9){
        println(i + "x" + j + "=" + i*j)
      }
    }
  }
}
```

# B   PARALLEL DATA CREATED FROM CODENET

In this section we look at the various properties of the parallel dataset created from the CodeNet dataset. In section B.1, we present most similar data samples identified by the WMD metric across various language pairs. In section B.2, we present the histograms of similarity scores in the final dataset. Finally, in section B.3, we present the number of data samples in the final dataset created from the CodeNet dataset.

## B.1   IDENTIFIED MOST SIMILAR DATA SAMPLES ACROSS LANGUAGE PAIRS

In table 4, we provide the most-similar code samples identified by the WMD metric across various language pairs along with the computed similarity between the two samples.

Table 4: Most similar code samples across various language pairs as identified by the WMD metric

| C → Python dataset: Similarity: 0.75 |
| --- |

```c
#include <stdio.h>
#include <math.h>

int main()
{
  int n;
  int i;
  double x1, x2, x3, y1, y2, y3, px, py, r;

  scanf("%d", &n);

  for(i = 0; i < n; i++){

    scanf("%lf %lf %lf %lf %lf %lf", &x1, &y1, &x2
      ↪ , &y2, &x3, &y3);

    px = ((y2 - y3)*(x1*x1 + y1*y1) + (y3 - y1)*(
      ↪ x2*x2 + y2*y2) + (y1 - y2)*(x3*x3 +
      ↪ y3*y3))/(2*(x1*(y2 - y3) + x2*(y3 -
      ↪ y1) + x3*(y1 - y2)));
    py = ((x2 - x3)*(x1*x1 + y1*y1) + (x3 - x1)*(
      ↪ x2*x2 + y2*y2) + (x1 - x2)*(x3*x3 +
      ↪ y3*y3))/(2*(y1*(x2 - x3) + y2*(x3 -
      ↪ x1) + y3*(x1 - x2)));

    r = sqrt(pow((x1 - px), 2) + pow((y1 - py), 2)
      ↪ );

    printf("%.3f %.3f %.3f\n", px, py, r);
  }

  return 0;
}
```

```python
import math

n = int(raw_input())

for i in range(n):

    x1, y1, x2, y2, x3, y3 = map(float, raw_input().
      ↪ split())

    px = ((y2 - y3)*(x1*x1 + y1*y1) + (y3 - y1)*(x2*
      ↪ x2 + y2*y2) + (y1 - y2)*(x3*x3 + y3*y3)
      ↪ )/(2*(x1*(y2 - y3) + x2*(y3 - y1) + x3
      ↪ *(y1 - y2)))
    py = ((x2 - x3)*(x1*x1 + y1*y1) + (x3 - x1)*(x2*
      ↪ x2 + y2*y2) + (x1 - x2)*(x3*x3 + y3*y3)
      ↪ )/(2*(y1*(x2 - x3) + y2*(x3 - x1) + y3
      ↪ *(x1 - x2)))

    r = math.sqrt(pow((x1 - px), 2) + pow((y1 - py),
      ↪ 2))

    print "%.3f %.3f %.3f" % (px, py, r)
```

| C# → Java dataset: Similarity: 0.62 |
| --- |

```csharp
using System;

class Program
{
    static void Main(string[] args)
    {
        for (int i = 0; i < 1000; i++)
        {
            Console.WriteLine("Hello World");
        }
    }
}
```

```java
class Main
{
    public static void main(String[] args)
    {
        for (int i = 0; i < 1000; i++)
        {
            System.out.println("Hello World");
        }
    }
}
```

| Scala → Ruby dataset: Similarity: 0.91 |
| --- |

```scala
object Main extends App {
    val a = Array(1, 1, 1, 2, 1, 2, 1, 5, 2, 2, 1,
      ↪ 5, 1, 2, 1, 14, 1, 5, 1, 5, 2, 2, 1,
      ↪ 15, 2, 2, 5, 4, 1, 4, 1, 51)
    val k = scala.io.StdIn.readInt - 1
    println(a(k))
}
```

```ruby
k = gets.to_i
a = "1, 1, 1, 2, 1, 2, 1, 5, 2, 2, 1, 5, 1, 2, 1,
  ↪ 14, 1, 5, 1, 5, 2, 2, 1, 15, 2, 2, 5, 4,
  ↪ 1, 4, 1, 51".split(",").map(&:to_i)
p a[k-1]
```

| PHP → Python dataset: Similarity: 1.34 |
| --- |

```php
<?php

while ($line = trim(fgets(STDIN))) {
    $num = explode(" ",$line);
    if ($num[0] == 0 && $num[1] == 0) {
        exit();
    }
    sort($num);
    $line = implode(" ",$num);
    echo $line . PHP_EOL;
}

?>
```

```python
while True:
    num = [int(x) for x in input().rstrip().split
        ↪ ()]
    if num[0] ==0 and num[1] == 0:
        break
    num.sort()
    ans = " ".join(str(x) for x in num)
    print(ans)

# <?php

# while ($line = trim(fgets(STDIN))) {
#     $num = explode(" ",$line);
#     if ($num[0] == 0 && $num[1] == 0) {
#         exit();
#     }
#     sort($num);
#     $line = implode(" ",$num);
#     echo $line . PHP_EOL;
# }

# ?>
```

## B.2 SIMILARITY HISTOGRAMS IN THE CREATED DATASET

We show the histograms of the similarity scores in the datasets matched through the WMD metric on the CodeNet dataset in figure 4.

## B.3 DATA STATISTICS

In table 5, we provide the number of data samples in the final dataset created from the CodeNet dataset.

Table 5: Number of data samples for each code translation dataset. Each row represents the source language, while each column represents the target language. Due to the skewed number of submissions in different languages in the CodeNet dataset, we see the same in the created parallel dataset.

|  | C | C# | C++ | Go | Java | JS | PHP | Python | Ruby | Scala |
|---|---|---|---|---|---|---|---|---|---|---|
| **C** | × | 2,295 | 15,534 | 709 | 8,900 | 1,354 | 837 | 10,499 | 5,564 | 1,247 |
| **C#** | 2,504 | × | 3,145 | 595 | 2,629 | 846 | 635 | 2,639 | 2,007 | 821 |
| **C++** | 30,977 | 3,388 | × | 1,358 | 17,809 | 1,793 | 1,550 | 44,707 | 14,375 | 2,183 |
| **Go** | 1,732 | 974 | 2,496 | × | 2,077 | 642 | 669 | 1,755 | 1,315 | 729 |
| **Java** | 15,909 | 5,016 | 23,821 | 2,858 | × | 2,262 | 2,202 | 12,846 | 7,746 | 2,034 |
| **JS** | 2,876 | 1,907 | 2,954 | 998 | 2,653 | × | 1,238 | 2,893 | 2,307 | 1,349 |
| **PHP** | 2,802 | 1,741 | 2,898 | 1,180 | 2,454 | 1,389 | × | 2,996 | 2,310 | 1,133 |
| **Python** | 27,947 | 7,778 | 76,825 | 7,235 | 29,007 | 6,666 | 7,641 | × | 48,301 | 7,126 |
| **Ruby** | 28,423 | 12,595 | 48,985 | 8,838 | 23,515 | 8,322 | 9,252 | 62,298 | × | 6,457 |
| **Scala** | 5,506 | 4,720 | 6,660 | 3,343 | 5,953 | 3,536 | 3,249 | 6,937 | 5,586 | × |

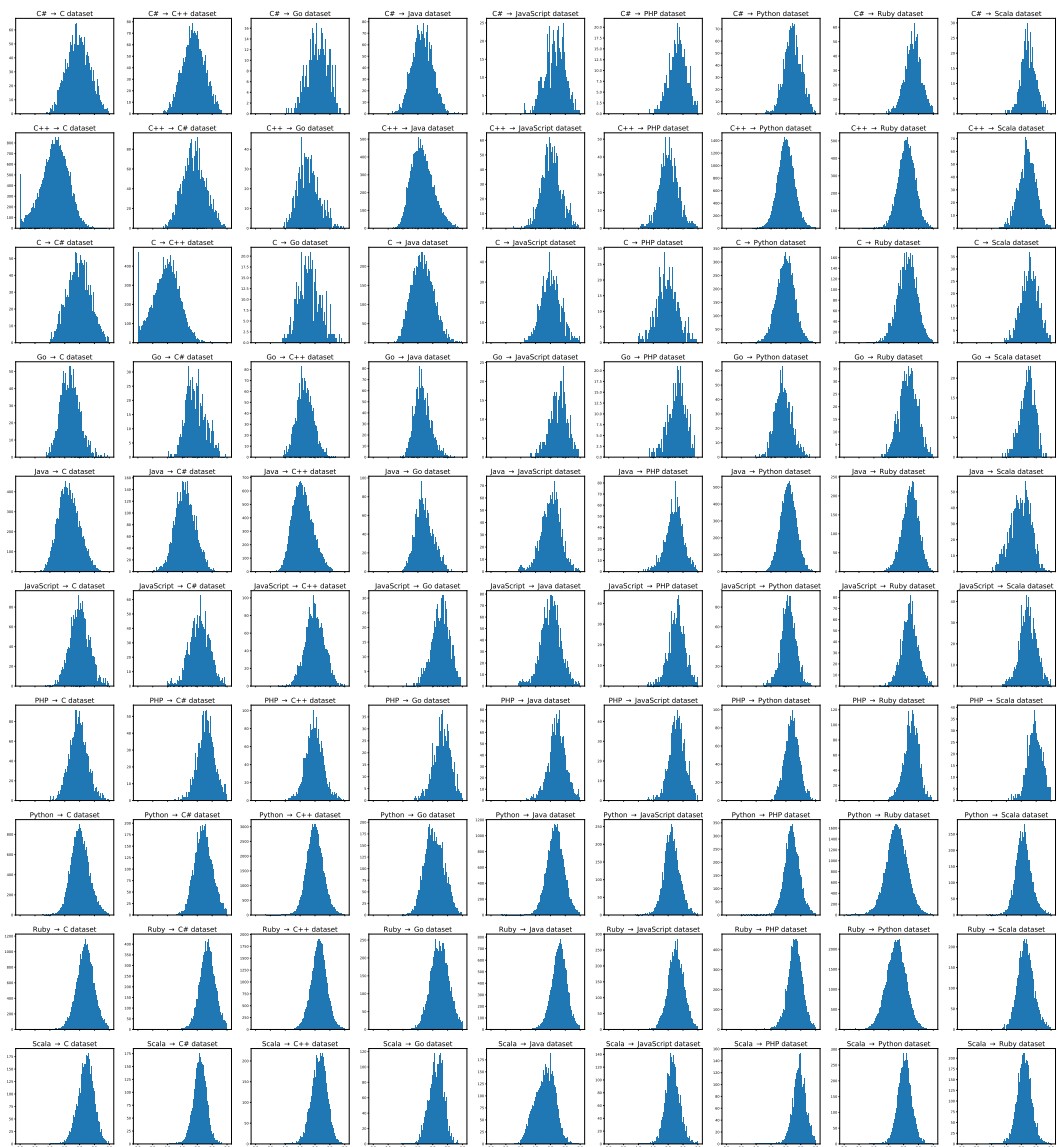

Figure 4: Histogram of similarity scores for various language pairs matched by WMD

### B.4 CODE GENERATED FROM FINE-TUNED PLBART MODEL

In table 6, we show examples of code generated from a PLBART model fine-tuned on a dataset created using the WMD metric. We show examples from C → Python, C# → Java, C++ → Ruby, Java → Ruby, JavaScript → C, PHP → Python, Python → C, Ruby → C, and Scala → Python language pairs. For a given input source code, we show an example of the correctly generated and incorrectly generated code.

Table 6: Examples of correct and incorrect translations given the source code to a PLBART model fine-tuned on a dataset matched using the WMD metric

| C → Python translation | | |
| --- | --- | --- |
| Source C code | Correct Python translation | Incorrect Python translation |

```c
#include <stdio.h>
int main(void) {
long n;
scanf("%ld", &n);
long a[n];
for (long i = 0; i < n; i++) {
scanf("%ld", &a[i]);
}
for (long i = 0; i < n; i++) {
if (a[i]%2 != 0) {
printf("first\n");
return 0;
}
}
printf("second\n");
return 0;
}
```

```python
n = int(input())
a = list(map(int, input().split()
    ↪ ))
for i in range(n):
    if a[i]%2 != 0:
        print("first")
        exit()
print("second")
```

```python
n = int(input())
a = list(map(int, input().split()
    ↪ ))
for i in range(n):
    if a[i]%2 == 0:
        print("first")
        exit()
print("second")
```

| C# → Java translation | | |
| --- | --- | --- |
| Source C# code | Correct Java translation | Incorrect Java translation |

```csharp
using System;
namespace Tenka1A
{
class Program
{
static void Main(string[] args)
{
string[] input = Console.
    ↪ ReadLine().Split(' ');
int A = int.Parse(input[0]);
int B = int.Parse(input[1]);
int C = int.Parse(input[2]);
bool flg = false;
if (A > B)
{
if (C > B&&A>C)
{
flg = true;
}else if (B>A) {
if (B>C&&C>A)
{
flg=true;
}
}
if (flg)
{
Console.WriteLine("Yes");
}
else {
Console.WriteLine("No");
}
}
}
}
}
```

```java
import java.util.Scanner;
public class Main {
public static void main(String[]
    ↪ args) {
Scanner sc = new Scanner(System.
    ↪ in);
int A = sc.nextInt();
int B = sc.nextInt();
int C = sc.nextInt();
boolean flg = false;
if (A > B) {
if (C > B && A>C) {
flg = true;
} else if (A < B) {
if (B>C && C>A) {
flg = true;
}
}
if (flg) {
System.out.println("Yes");
} else {
System.out.println("No");
}
}
}
```

```java
import java.util.Scanner;
public class Main {
public static void main(String[]
    ↪ args) {
Scanner sc = new Scanner(System.
    ↪ in);
int A = sc.nextInt();
int B = sc.nextInt();
int C = sc.nextInt();
boolean flg = false;
if (A > B) {
if (C > B && A>C) {
flg = true;
} else if (A < B) {
if (B>C && A>C) {
flg = true;
}
}
if (flg) {
System.out.println("Yes");
} else {
System.out.println("No");
}
sc.close();
}
}
```

| C++ → Ruby translation | | |
| --- | --- | --- |
| Source C++ code | Correct Ruby translation | Incorrect Ruby translation |

```cpp
#include <cstdio>
#include <iostream>
#include <algorithm>
#include <cstring>
#include <cmath>
using namespace std;
typedef long long LL;
int n,k,ans = 0,flg = 0;
int a[500005];
int mx[500005] = {0};
int main(){
ios::sync_with_stdio(false);
cin >> n;
memset(mx,-0x3f,sizeof(mx));
mx[0] = 0;
for(int i = 1;i <= n;i ++) cin
     ↪ >> a[i];
for(int i = 1;i <= n;i ++){
if(a[i] == 1) flg = 1;
mx[a[i]] = max(mx[a[i]],mx[a[i]
     ↪ - 1] + 1);
ans = max(ans,mx[a[i]]);
}
if(!flg) cout << -1 << endl;
else cout << n - ans << endl;
return 0;
}
```

```ruby
N = gets.to_i
as = gets.split.map(&:to_i)
mx = Array.new(N + 1, -Float::
     ↪ INFINITY)
mx[0] = 0
as.each do |a|
if a == 1
mx[a] = 1
elsif mx[a] < mx[a - 1]
mx[a] = mx[a - 1] + 1
end
end
puts N - mx.max
```

```ruby
N = gets.to_i
as = gets.split.map(&:to_i)
mx = Array.new(N + 1, -Float::
     ↪ INFINITY)
mx[0] = 0
as.each do |a|
if a == 1
mx[a] = 1
elsif mx[a] < mx[a - 1]
mx[a] = mx[a - 1] + 1
end
end
ans = N - (mx.max + 1)
puts ans
```

| Java → Ruby translation | | |
| --- | --- | --- |
| Source Java code | Correct Ruby translation | Incorrect Ruby translation |

```java
import java.util.HashMap;
import java.util.Map;
import java.util.Scanner;
public class Main {
public static void main(String[]
     ↪ args) throws Exception
     ↪ {
// File file = new File("test.
     ↪ txt");
// Scanner sc = new Scanner(file
     ↪ );
Scanner sc = new Scanner(System.
     ↪ in);
int n = sc.nextInt();
Map<Integer, Boolean> map = new
     ↪ HashMap<>();
for(int i = 0; i < n; i++){
int a = sc.nextInt();
if(null != map.get(a)){
System.out.println("NO");
sc.close();
return;
}else{
map.put(a, true);;
}
}
sc.close();
System.out.println("YES");
}
}
```

```ruby
n = gets.to_i
a = gets.split.map(&:to_i)
h = Hash.new(0)
a.each do |i|
h[i] += 1
end
puts h.values.all? { |v| v == 1 }
     ↪ ? "YES" : "NO"
```

```ruby
n = gets.to_i
a = gets.split.map(&:to_i)
h = Hash.new(0)
a.each do |i|
h[i] += 1
end
puts h.values.all? { |v| v == 0 }
     ↪ ? "YES" : "NO"
```

| Javascript → C translation | | |
| --- | --- | --- |
| Source JavaScript code | Correct C translation | Incorrect C translation |

| | | |
|---|---|---|
| ```'use strict'
const main = input => {
const args = input.split('\n').
    → map(arg => arg.split('
    → '))
const A = parseInt(args[0][0],
    → 10)
const B = parseInt(args[0][1],
    → 10)
if (A % 3 === 0 || B % 3 === 0
    → || (A + B) % 3 === 0)
    → console.log('Possible')
else console.log('Impossible')
}
main(require('fs').readFileSync
    → ('/dev/stdin', 'utf8'))
``` | ```#include <stdio.h>
int main(void)
{
int a, b;
scanf("%d %d", &a, &b);
if (a % 3 == 0 || b % 3 == 0 || (
    → a + b) % 3 == 0) printf(
    → "Possible\n");
else printf("Impossible\n");
return 0;
}
``` | ```#include <stdio.h>
int main(void)
{
int a, b;
scanf("%d %d", &a, &b);
if (a % 3 == 0 || b % 3 == 0)
printf("Possible\n");
else
printf("Impossible\n");
return 0;
}
``` |

**PHP → Python translation**

| Source PHP code | Correct Python translation | Incorrect Python translation |
|---|---|---|
| ```<?php

fscanf(STDIN,"%s",$s);$flg =
    → false;
if(strpos($s, "A") === false){
}else{
if(strpos($s,"B") === false){
}else{
$flg = true;
}
}

if($flg){
echo "Yes";
}else{
echo "No";
}
``` | ```s = input()
flg = False
if "A" in s or "B" in s:
    flg = True
else:
    if "A" in s:
        flg = True
print("Yes" if flg else "No")
``` | ```s = input()
a = 0
b = 0
for i in range(len(s)):
    if "A" in s[i]:
        if "B" in s[i]:
            a += 1
        else:
            b += 1
if a == b:
    print("Yes")
else:
    print("No")
``` |

**Python → C translation**

| Source Python code | Correct C translation | Incorrect C translation |
|---|---|---|
| ```import sys
N = int(input())
a = list(map(int, input().split
    → ()))
ans = 0
count = 1
if 1 not in a :
    print(-1)
    sys.exit()
for i in range(N) :
    if a[i] == count :
        count += 1
    else :
        ans += 1
print(ans)
``` | ```#include <stdio.h>
int main()
{
int n;
scanf("%d", &n);
int i;
int a[100005];
for (i = 0; i < n; i++)
scanf("%d", &a[i]);
int ans = 0;
int count = 1;
for (i = 0; i < n; i++)
{
if (a[i] == count)
count++;
else
ans++;
}
printf("%d\n", ans);
return 0;
}
``` | ```#include <stdio.h>
int main(void){
int N;
scanf("%d", &N);
int a[N];
for (int i = 0; i < N; i++) {
scanf("%d", &a[i]);
}
int ans = 0;
int count = 1;
for (int i = 0; i < N; i++) {
if (a[i] == count) {
ans++;
count++;
}
}
printf("%d\n", ans);
return 0;
}
``` |

**Ruby → C translation**

| Source Ruby code | Correct C translation | Incorrect C translation |
|---|---|---|

```ruby
A,B = gets.split.map(&:to_i)
S = gets.chomp
if /^\d{#{A}}-\d{#{B}}/.match(S)
puts 'Yes'
else
puts 'No'
end
```

```c
#include<stdio.h>
int main(void)
{
int A,B;
scanf("%d %d",&A,&B);
char S[100];
scanf("%s",S);
if(S[0]=='A'&&S[1]=='B'&&S[2]=='A
    ↪ '){
printf("No");
return 0;
}
if(S[0]=='A'&&S[1]=='B'&&S[2]=='A
    ↪ '){
printf("No");
return 0;
}
if(S[0]=='A'&&S[1]=='B'&&S[2]=='A
    ↪ '){
printf("No");
return 0;
}
if(S[0]=='A'&&S[1]=='B'&&S[2]=='A
    ↪ '){
printf("No");
return 0;
}
if(S[0]=='A'&&S[1]=='B'&&S[2]=='A
    ↪ '){
printf("No");
return 0;
}
if(S[0]=='A'&&S[1]=='B'&&S[2]=='A
    ↪ '){
printf("No");
return 0;
}
if(S[0]=='A'&&S[1]=='B'&&S[2]=='A
    ↪ '){
printf("No");
return 0;
}
if(S[0]=='A'&&S[1]=='B'&&S[2]=='B
    ↪ '){
printf("No");
return 0;
}
printf("Yes");
return 0;
}
```

```c
#include<stdio.h>
int main(void)
{
int A,B;
scanf("%d %d",&A,&B);
char S[100];
scanf("%s",S);
if(S[0]=='A'&&S[1]=='B'&&S[2]=='A
    ↪ '&&S[3]=='B'&&S[4]=='A'
    ↪ &&S[5]=='B'&&S[6]=='A'){
printf("Yes\n");
}
else{
printf("No\n");
}
return 0;
}
```

| Scala → Python translation | | |
| --- | --- | --- |
| Source Scala code | Correct Python translation | Incorrect Python translation |

```scala
import scala.io.Source
object Main extends App {
val lines: Iterator[String] =
    ↪ Source.stdin.getLines()
val line = lines.next.split(" ")
    ↪ .map(_.toInt).take(2)
val a = line.head
val b = line(1)
println((if (b >= a) a else a -
    ↪ 1).toString)
}
```

```python
import sys
lines = sys.stdin.readlines()
for line in lines:
    a, b = map(int, line.split()
        ↪ )
    if b >= a:
        print(a)
    else:
        print(b - 1)
```

```python
import sys
import os
f = lambda:list(map(int,input().
    ↪ split()))
if 'local' in os.environ :
    sys.stdin = open('./input.
        ↪ txt', 'r')
def solve():
    a = f()[0]
    b = f()[0]
    print(a if b >= a else a -
        ↪ 1)
solve()
```

