# OpenReview forum: "Using Document Similarity Methods to create Parallel Datasets for Code Translation"
_ICLR.cc/2022/Conference — ICLR 2022 Submitted_

### Official Review · Reviewer_3acE · 2021-11-01

**Correctness:** 4
**Technical Novelty And Significance:** 2
**Empirical Novelty And Significance:** 2
**Recommendation:** 3
**Confidence:** 4

**Main Review:**

In its current form, the problem statement and approach seem very trivial. First, the data sets seem rather unsophisticated and it is not clear what the pathway is from this method to say mining parallel corpora from github or some large-scale code repository websites. In its current form, it seems like this is preliminary work and with some more sophistication (in method + data sets) it is going to be a good future task. I suggest that this be submitted to an appropriate workshop and a future attempt be made to submit this to ICLR. In particular, good directions can include in-the-wild performance when tested on something outside of coding-problem data sets (especially since those are so well-organized by language and problem).



**Summary Of The Paper:**

The paper constructs (weak) parallel corpora for code translation tasks using nearest neighbor sampling in the code-document-embedding space. A variety of similarity metrics / base LMs are evaluated.

**Summary Of The Review:**

The method and data set in this paper in its current form are rather trivial and more sophistication is required. The techniques seem highly tuned for mining well-organized data sets from coding-problem websites. A better and more improved model that can scale in the wild is recommended and would be a better fit for this venue.

---

> ### Author Response · Authors · 2021-11-15
> **Author Response to Reviewer 3acE**
>
> We sincerely thank the reviewer for taking the time out to review our paper. Here, we aim to address the concerns that you've raised in your review:
>
> 1. `In its current form, the problem statement and approach seem very trivial`
>     1. We would like to very strongly argue that the problem statement is not trivial at all. Research in using ML techniques for Source Code Translation has been ongoing since 2013, but as of today, there are only 2 datasets available for the task. All prior works in this space have either used the Java ↔ C# dataset or existing transpiler tools such as Python 2 to 3 translator to create datasets. We would like to quote from the TransCoder paper [1] to further the point that the problem statement is not trivial, and has hampered progress in this field for long:
>
>     > Although neural models significantly outperform their rule-based counterparts in the context of natural language translation, their applications to transcompilation have been limited due to the scarcity of parallel data in this domain.
>
>     2. We argue that the proposed approach is simple and not trivial. We very strongly believe that unnecessary sophistication should not be a prerequisite to submission at a research conference. Our approach:
>
>     - has no precedence in literature and is hence novel,
>     - is simple to understand and use, and
>     - we thoroughly demonstrate its effectiveness and back up our claims through thorough experiments as noted by reviewers nRDP and xtd8.
>
>     We, therefore, respectfully believe that approach being trivial is not true and it being simple should not be a valid criticism of the work.
>
> 2. `not clear what the pathway is from this method to say mining parallel corpora from github`
>     1. We would like to kindly refer the reviewer to the Parallel Corpus Mining paragraph in the Related Work section of the paper. There, we cite similar prior works in the natural language domain, some of which have applied techniques of the same nature as we've proposed to bigger datasets and corpus. Specifically, we refer the works of
>         - Banón et al [2], who use TF-IDF and distance between embeddings to mine parallel corpora from over 150000 websites
>         - Uszkoreit et al [3], who use n-gram features to mine parallel corpora from 2.5 billion web pages
>         - Artetxe and Holger [4], who use multilingual sentence embeddings to mine parallel corpora from 64.4 million sentence pairs
>
>     2. Referring to these papers and broadly referring to prior literature in the NL domain, it should be pretty clear on scaling the proposed technique to large-scale corpora.
>
> 3. `in-the-wild performance when tested on something outside of coding-problem data sets`
>     1. In principle, we agree that evaluation of code translation models should be on a broader set of problems. However, if you look at the state of research in this domain, you will find that it is a standard way of evaluating code generation models on competitive programming type problem sets, like we do. We refer the reviewer to:
>         1. Chen et al. [5], who proposed OpenAI Codex, and evaluated the model on 164 programming problems they call the HumanEval dataset
>         2. Hendrycks et al. [6], who assessed the performance of GPT models on this dataset on 10,000 programming problems dataset
>         3. Lachaux et al. [1] who evaluate their model on 400 programming problems, and
>         4. Austin et al. [7], who evaluate large language models on basic programming problems
>
>     2. We follow the standard way of evaluating code generation and code translation models, as has been followed in prior works. Evaluating models for code on samples outside of coding-problem data is out-of-scope for this work, and can be part of a future work in this domain.
>
> We hope our rebuttal address your concerns. If not, we would be very happy to engage with you further. We look forward to having a fruitful discussion with you.
>
>
>
> [1] Lachaux, Marie-Anne, et al. "Unsupervised translation of programming languages." *arXiv preprint arXiv:2006.03511* (2020).
>
> [2] Banón, Marta, et al. "ParaCrawl: Web-scale acquisition of parallel corpora." *Proceedings of the 58th Annual Meeting of the Association for Computational Linguistics*. 2020.
>
> [3] Uszkoreit, Jakob, et al. "Large scale parallel document mining for machine translation." *Proceedings of the 23rd International Conference on Computational Linguistics (Coling 2010)*. 2010.
>
> [4] Artetxe, Mikel, and Holger Schwenk. "Margin-based parallel corpus mining with multilingual sentence embeddings." *arXiv preprint arXiv:1811.01136* (2018).
>
> [5] Chen, Mark, et al. "Evaluating large language models trained on code." *arXiv preprint arXiv:2107.03374* (2021).
>
> [6] Hendrycks, Dan, et al. "Measuring Coding Challenge Competence With APPS." *arXiv preprint arXiv:2105.09938* (2021).
>
> [7] Austin, Jacob, et al. "Program synthesis with large language models." *arXiv preprint arXiv:2108.07732* (2021).

---

### Official Review · Reviewer_rSze · 2021-11-02

**Correctness:** 2
**Technical Novelty And Significance:** 2
**Empirical Novelty And Significance:** 2
**Recommendation:** 3
**Confidence:** 4

**Main Review:**

Although I recognized that the main contribution of this paper is not proposing a sophisticated method to align samples, I thought the paper need more attention to design efficient strategy. The entire alignment algorithm takes O(D*f(D')) where f(D') is the complexity of GetSimilarDocuments(). Unless any assumption this takes O(DD') meaning that it tends to be infeasible by increasing the data. Also, the algorithm seems a simple greedy method and may be heavily affected by the sampling order.

According to Table 2 and 3, I observed that the pseudo-match accuracy introduced here does not reasonably reflect the real characteristics needed to be considered in code translation models since the BLEU of several systems compete with each other.

I recognized that Figure 1 shows every range of noise ratio, except around 100%, has certain gradient against the accuracy, and got the opposite conclusion "the less noise the better" against that the paper concluded "we expect a certain amount of noise, we can expect the models to perform reasonably well." The root cause of this is that the paper did not provide a reasonable criterion about acceptance of this experiment first.

As far as I saw the Figure 2 and 3, the accuracy is heavily reflected by the amount data available during training and some normalization should be needed to compare these metrics each other without biases. Figure 3 also show some remarkable characteristics in translating only into C++, but the paper did not mention about this point.

The motivation of the paper includes "modernizing legacy applications," but the experiments does not reflect this perspective. For example, the first section repeatedly referred to the COBOL applications, but we maybe not be able to reflect the experiments into this application due to large discrepancy between languages.

**Summary Of The Paper:**

This paper proposes to use similarity metric to generate pseudo alignments between source/target program pairs. Generated pairs are utilized to train program translation model. The paper described a simple greedy method to align both codes, and experimented 5 types of similarity metric as its inner measure.
According to the experiments, the word mover's distance works notably well for this purpose, but other metric can also improve the translation accuracy significantly against random selection.
The other experiment investigating a performance curve by changing noise ratio in the ground-truth parallel corpus hypothesized that certain amount of alignment errors can be acceptable since the actual performance can be maintained.
The paper also challenged to construct translation systems between arbitrary pairs in 10 programming languages using the proposed framework and observed that the trained system works with certain accuracies.

**Summary Of The Review:**

- The alignment algorithm need to be improved. This is not critical: this is still acceptable for the first observation.
- Conclusion sounds sometimes not reasonable according to experiments. Some results are hard to discuss due to unnormalized data.

---

> ### Author Response · Authors · 2021-11-15
> **Author Response to Reviewer rSze - Part 1**
>
> 1. `the paper need more attention to design efficient strategy`
>     1. We acknowledge that the proposed algorithm, as it stands, is quadratic in nature, and reducing its time complexity is desirable. However, we would like to emphasize, that even with its quadratic nature the algorithm is still tractable and applicable to large-scale corpora. Prior works in the natural language have applied similar strategies (with similar asymptotic time complexity) to align documents for large-scale corpora. Specifically, we refer the reviewer to the works of:
>         1. Banón et al [1], who mine parallel corpora from over 150,000 websites to ultimately create over 223 million (post-filtration) parallel natural language samples. The authors here use TF-IDF and cosine distance between the document embeddings as two methods to mine parallel corpora.
>         2. Uszkoreit et al [2], who use n-gram features to mine parallel corpora from 2.5 billion web pages
>         3. Artetxe and Holger [3], who use multilingual sentence embeddings to mine parallel corpora from 64.4 million sentence pairs
>     2. Therefore, our technique, even in its current form, can be applied to large-scale corpora as [1,2,3] demonstrate.
> 2. `According to Table 2 and 3, I observed that the pseudo-match accuracy introduced here does not reasonably reflect the real characteristics needed to be considered in code translation models since the BLEU of several systems compete with each other.`
>     1. The pseudo-match accuracy used in Table 2 is not meant to evaluate code translation models, but to measure the effectiveness of document similarity metrics to create parallel documents. This metric is a relaxation of the Match Accuracy metric used in Table 1 experiments and is introduced for Table 2 experiments because the CodeNet dataset does not have parallel code samples available to enable us to compute the Match Accuracy.
>     2. Additionally, Table 1 contains the corresponding Match Accuracy results for Table 3 results, and these two tables need to be read together.
> 3. `Figure 1 shows every range of noise ratio, except around 100%, has certain gradient against the accuracy, and got the opposite conclusion "the less noise the better"`
>     1. While the mean CodeBLEU score does increase slightly when going from 90% noise to 100% noise, its variance also increases. Additionally, all the other metrics, for all the other noise levels, and both ways of translation, follow the pattern supporting our conclusion that "less noise the better". We believe, that increase in the mean CodeBLEU score but also an increase in the variance values, are a random-effect and insufficient to refute the conclusion we draw.
>     2. `The root cause of this is that the paper did not provide a reasonable criterion about acceptance of this experiment first.`
>         1. We're not completely sure on what the reviewer meant by "reasonable criterion about acceptance of this experiment". We interpreted it two ways, and provide our explanations to both lines of thought below. But, we would really appreciate if the reviewer could clarify this point.
>             1. Thought 1: Acceptance of the experiment refers to the motivation behind this experiment and why is it needed: In Table 1 and Table 2 experiments, we note that the datasets created using our proposed method will be noisy, and therefore, we intend to study the noise vs performance characteristics of models trained on code. Figure 1 experiment, therefore, aims to quantify the effect noise has on model performance for various levels of noise.
>             2. Thought 2: Acceptance of the experiment refers to an acceptable level of evaluation metric score vs noise: As we note above that the motivation of this experiment is to demonstrate the variation in model performance with increasing levels of noise. For this purpose, we show the complete range of noise starting from 0% (no noise) to 100% (complete random pairings), as a way to measure the model performance for the complete range of noise.
>
>
> [1] Banón, Marta, et al. "ParaCrawl: Web-scale acquisition of parallel corpora." *Proceedings of the 58th Annual Meeting of the Association for Computational Linguistics*. 2020.
>
> [2] Uszkoreit, Jakob, et al. "Large scale parallel document mining for machine translation." *Proceedings of the 23rd International Conference on Computational Linguistics (Coling 2010)*. 2010.
>
> [3] Artetxe, Mikel, and Holger Schwenk. "Margin-based parallel corpus mining with multilingual sentence embeddings." *arXiv preprint arXiv:1811.01136* (2018).

---

> ### Author Response · Authors · 2021-11-15
> **Author Response to Reviewer rSze - Part 2**
>
> 4. `accuracy is heavily reflected by the amount data available during training and some normalization should be needed to compare these metrics each other without biases`
>     1. We would like to emphasize that performance is meant to be compared between Figure 2 and Figure 3, where Figure 2 is the score when WMD is used to create the parallel dataset and Figure 3 is the score when random matching is used to create the parallel dataset.
>     2. For these two sets of metrics (Figure 2 and Figure 3), all the experimental settings are kept the same. For example, the number of data samples, problem sets used for training and testing, model hyperparameters are the same between corresponding values of Figure 2 and Figure 3. These are done to ensure we can compare the metrics with one another.
>     3. Additionally, metrics within Figure 2 are not meant to be compared to each other because:
>         1. As the reviewer noted, there is a difference in the number of data samples. This is a natural effect of the popularity of different programming languages. Some languages such as C, C++ are much more popular than other languages and therefore contain more code samples than others.
>         2. The programming language's semantic also play an important role in the model performance. For example: translating from a dynamically typed language such as Python to a statically typed language such as Java is a harder problem since the model now needs to infer the variable data type as well. This phenomenon is also observed in the works of Lachaux et al. [4] (see Table 2)
>         3. For these reasons, we believe that Figure 2 results are not meant to be compared to each other, but rather to the corresponding results in Figure 3.
> 5. `motivation of the paper includes "modernizing legacy applications," but the experiments does not reflect this perspective`
>     1. While we motivate from the case of translating from COBOL to a modern language alone, there are other instances of modernization, some of which we cover in our experiments. For example, Etsy modernized its codebase from JavaScript to TypeScript [5], and Khan Academy modernized its codebase from Python to Go [6]. In our experiments, we cover the Python to Go translation (see Figure 2), along with other sets of translation from older languages such as from PHP to modern languages such as Ruby.
>     2. We will update the paper to reflect these new use-cases and bring forth the point more clearly that code translation need not be limited to modernizing legacy applications but can also be practically applicable for translation between relatively modern languages.
>
>
>     [1] Banón, Marta, et al. "ParaCrawl: Web-scale acquisition of parallel corpora." *Proceedings of the 58th Annual Meeting of the Association for Computational Linguistics*. 2020.
>
>     [2] Uszkoreit, Jakob, et al. "Large scale parallel document mining for machine translation." *Proceedings of the 23rd International Conference on Computational Linguistics (Coling 2010)*. 2010.
>
>     [3] Artetxe, Mikel, and Holger Schwenk. "Margin-based parallel corpus mining with multilingual sentence embeddings." *arXiv preprint arXiv:1811.01136* (2018).
>
>     [4] Lachaux, Marie-Anne, et al. "Unsupervised translation of programming languages." *arXiv preprint arXiv:2006.03511* (2020).
>
>     [5] [https://codeascraft.com/2021/11/08/etsys-journey-to-typescript/](https://codeascraft.com/2021/11/08/etsys-journey-to-typescript/)
>
>     [6] [https://blog.khanacademy.org/go-services-one-goliath-project/](https://blog.khanacademy.org/go-services-one-goliath-project/)

---

### Official Review · Reviewer_xtd8 · 2021-11-04

**Correctness:** 3
**Technical Novelty And Significance:** 2
**Empirical Novelty And Significance:** 3
**Recommendation:** 5
**Confidence:** 3

**Main Review:**

**Strengths**
1. The paper is easy to follow. Although I am not super familiar with code translation, I can catch most of the points.
2. The proposed method is simple and easy to use.
3. The experiments are relatively thorough, covering different types of document similarity methods, programming languages, model architectures, and evaluation metrics. The results seem to be convincing.

**Weaknesses**
1. My main doubt is the results presented in "Section 5.3 RQ3: TRANSLATING BETWEEN A WIDER SET OF PROGRAMMING LANGUAGES". The authors report CA@5 scores here but in the other place, BLEU/CodeBLEU/EM is used. So, what is the intuition that uses different metrics? BLEU can also handle multi-reference evaluation. It would be nice if the authors could add these results in the author response.
2. The research is somewhat superficial. The authors only show the models can tolerate certain noises but do not propose any simple heuristics to alleviate the impact of noises. For example, penalizing the noises (i.e., the targets not belonging to the source) during model training (fine-tuning). [1] might inspire the authors.

[1] Wu, Lijun, Jinhua Zhu, Di He, Fei Gao, Tao Qin, Jianhuang Lai, and Tie-Yan Liu. "Machine translation with weakly paired documents." In Proceedings of the 2019 Conference on Empirical Methods in Natural Language Processing and the 9th International Joint Conference on Natural Language Processing (EMNLP-IJCNLP), pp. 4375-4384. 2019.

**Typos**
Section 3 Proposed Method: curated -> created

**Summary Of The Paper:**

This paper mines noisy parallel datasets of code by calculating the similarity between two non-parallel sets of documents. The authors first show that the document similarity methods can indeed align parallel documents and find that the word movers distance (WMD) is the most effective one. Then, the authors show the high tolerance of models trained with noisy datasets. Based on the two findings, the authors finally apply the proposed method to a large, non-parallel code dataset, and observe a performance boost of using a noisy parallel dataset compared to randomly paired datasets.

**Summary Of The Review:**

This paper is interesting, but the research is somewhat superficial.

---

> ### Author Response · Authors · 2021-11-15
> **Author Response to Reviewer xtd8**
>
> We thank the Reviewer for taking the time out to review our paper and provide constructive feedback to our work. We're happy that you found the paper easy to understand and our experiments thorough. We now aim to address the concerns that you've raised:
>
> 1. `what is the intuition that uses different metrics?`
>     1. CA@k (k=5 in our experiments) is a recently introduced metric that's now the standard way of evaluating code generation and translation models [1, 2, 3]. However, evaluating this metric requires the availability of expected input-output pairs for the code samples in the test data.
>     2. For experiments in Table 3 and Figure 1 (where we use BLEU, CodeBLEU, and EM metrics), we are compelled to use Java ↔ C# dataset, since we need to compare the performance with the ground-truth performance, and therefore we need a dataset with ground truth parallel code samples available. Java ↔ C# is the only such dataset available.
>     3. We cannot compute the CA@k metric on the Java ↔ C# dataset because computing CA@k requires the availability of expected input-output pairs, which are not available for the Java ↔ C# dataset. We, therefore, use the 3 metrics that were used to evaluate code generation/translation systems prior to CA@k. We report results on all 3 metrics to provide a holistic summary of the results.
>     4. For RQ3, we evaluate the CA@k metric since (a) CodeNet provides us with expected input-output pairs enabling us to evaluate CA@k, and (b) it is now the standard metric of evaluating, interpreting, and comparing code generation/translation models.
>     5. While technically BLEU score can be computed for RQ3 experiments (as the reviewer noted about multi-reference evaluation), we do not report it since it has been shown to not be a good metric to measure code synthesis — please refer to Section 2 in Ren et al. [2], and Section 4.7 in Austin et al. [5].
> 2. `authors only show the models can tolerate certain noises but do not propose any simple heuristics to alleviate the impact of noises`
>     1. Here, we would like to emphasize that the primary contribution of this work is to show that document similarity methods can be used to create parallel datasets for code, which otherwise, have been unavailable to the research community beyond 2 datasets and programming languages. Since the datasets created by our proposed technique are bound to be noisy, we study the noise characteristics of models to show that these models can tolerate certain amounts of noise and therefore our technique has practical applicability.
>     Prior to our work, there was no concept of noisy datasets for code translation, and having now introduced this notion, future work might look at specifically dealing with noise in the dataset and draw inspiration from works in the natural language domain much like the one the reviewer referred us to. We, therefore, believe that dealing with noisy datasets in the code translation is a good natural future step, but outside of the scope of this work.
>
> [1] Lachaux, Marie-Anne, et al. "Unsupervised translation of programming languages." *arXiv preprint arXiv:2006.03511* (2020).
>
> [2] Chen, Mark, et al. "Evaluating large language models trained on code." *arXiv preprint arXiv:2107.03374* (2021).
>
> [3] Hendrycks, Dan, et al. "Measuring Coding Challenge Competence With APPS." *arXiv preprint arXiv:2105.09938* (2021).
>
> [4] Ren, Shuo, et al. "Codebleu: a method for automatic evaluation of code synthesis." *arXiv preprint arXiv:2009.10297* (2020).
>
> [5] Austin, Jacob, et al. "Program synthesis with large language models." *arXiv preprint arXiv:2108.07732* (2021).

---

### Official Review · Reviewer_nRDP · 2021-11-06

**Correctness:** 4
**Technical Novelty And Significance:** 2
**Empirical Novelty And Significance:** 2
**Recommendation:** 5
**Confidence:** 5

**Main Review:**

+ The proposed technique can be useful for legacy code translation where annotated data is scarce.
+ The empirical investigations are solid.
+ The paper is well-written and easy to understand.

- The authors did not provide any explanation/intuition/evidence why such a simple data augmentation technique works.

- The main problem with this approach is that the performance suffers significantly with CodeNet dataset. The reason behind similarity-based data augmentation will work is because for implementing similar functionalities often similar variable names are used. However, such an assumption may not be true for CodNet where developers did independent implementations. Thus, the dropping of performance for CodeNet is not surprising.  However, this raises questions about the usability of this technique.

- The contrast between performance with noise is useful, but again that shows how sensitive this technique is with noise and for some legacy code (e.g., COBOL --- a language authors use to motivate the study) the amount of noise might be high.




**Summary Of The Paper:**

The paper proposes a simple similarity-based data augmentation technique to translate code written in one programming language to another. With empirical investigation, they show such apparently simple data augmentation closely matches the performance of the models trained on the manually annotated datasets.

**Summary Of The Review:**

The authors propose an approach of data augmentation for code translation for document similarities. However, the results show the approach suffered for independently developed code (CodeNet), which is the most realistic case. The approach is also susceptible to noise.

---

> ### Author Response · Authors · 2021-11-15
> **Author Response to Reviewer nRDP - Part 1**
>
> We thank the Reviewer for taking the time out to review our paper and provide constructive feedback to our work. We're happy that you found the paper well-written and our empirical investigations thorough. We now wish to address the concerns that you've raised:
>
> 1. `The authors did not provide any explanation/intuition/evidence why such a simple data augmentation technique works`
>     1. First, we would like to mention that we propose and experiment with settings where no prior datasets exist and therefore we create the entire dataset using the proposed technique alone. Therefore, while the technique would work as a data augmentation approach, it is primarily experimented with as a dataset creation technique in our work.
>     2. The three main reasons why this simple technique works are:
>         1. Availability and ease of training code embeddings such as available through PLBART and GraphCodeBERT models. Code embeddings are trained in an unsupervised manner, and therefore can be trained on easily available data such as from GitHub. Additionally, these embeddings have been shown to map similar keywords in multiple programming languages close to each other in the embedding space (see Figure 5 in [1] for an example).
>         2. Availability of document similarity metrics such as the WMD score, which operates in the embedding space and relies on similar entities mapped close to each other in the embedding space. Having these pre-requisite conditions available, WMD is then able to match documents even when they have no token overlapping, thus helping us to create parallel datasets for code.
>         3. Source code has a stricter syntax than natural language, thereby allowing models trained on code to bear moderate amounts of noise in the datasets.
>         4. Our proposed technique takes advantage of these three conditions, which are easily met for the code translation domain, to work well for the use case.
>     3. We will add this explanation in the updated version of the paper.
> 2. `The main problem with this approach is that the performance suffers significantly with CodeNet dataset. The reason behind similarity-based data augmentation will work is because for implementing similar functionalities often similar variable names are used`
>     1. Here, we would like to highlight that while Bag-of-Words (BOW) based similarity methods such as TF-IDF and BM25 work by matching similar tokens, Word Mover's Distance operates exclusively in the embedding space. Therefore, as long as similar tokens (similar variables and function names, similar keywords in different languages) are mapped to nearby spaces in the embedding space, WMD works well. Finding similar documents even when no tokens overlap in the two documents is a key motivation of the WMD method and is highlighted in Figure 1 of Kusner et al. [2]. In this figure, similarity is computed between the two phrases, "Obama speaks to the media in Illinois" and "The President greets the press in Chicago". Since there are no overlapping words in the two sentences, BOW based similarity methods fail to work well here, while WMD correctly captures the similarity since the words "Obama" and "President", and "Chicago" and "Illinois" are mapped close to each other in the embedding space.
>     2. We also present some examples from the dataset created for RQ3 experiments here, showing that the method does not necessarily rely on similar variables and function names to find similar samples.
>         1. Sample 1, for instance, shows the second-closest data sample found in the Scala to Go dataset. Here, notice that the variable names `start`, `end`, and `th` are found in the Scala code but not in the Go code. Similarly, the function `calcuPoint` exists only in the Scala code. Inversely, there are line comments in the Go code which are missing in the Scala code.
>         2. Similarly, in Sample 2 we show the closest sample between COBOL and Python. We would like to clarify here that we applied the proposed technique to COBOL and Python examples in the CodeNet dataset to get a representative example for the rebuttal. We do not experiment with COBOL in the paper, because the number of samples available in the CodeNet dataset is not enough to train or fine-tune a model. In Sample 2, besides the `n` and `m` variables, none of the other variable names match. Also, many of the keywords are different between the languages. Still, WMD is able to match the two code samples, without over-relying on the variable and method names.

---

> ### Author Response · Authors · 2021-11-15
> **Author Response to Reviewer nRDP - Part 2**
>
> 3. `shows how sensitive this technique is with noise and for some legacy code (e.g., COBOL --- a language authors use to motivate the study) the amount of noise might be high.`
>     1. We argue that our findings from Figure 1 suggest that models can tolerate moderate amounts of noise and still perform decently well. With higher levels of noise, such as over 30% value in our experiments, the performance starts degrading severely. In our RQ3 experiments (see Figure 2), we show significant competence in translating from a relatively older language such as PHP to more modern languages such as Java, Python, C++, and Ruby. We expect similar results to port over to translation from COBOL or other legacy languages, contingent on the availability of good cross-lingual embeddings that map similar keywords close by from both the legacy and the modern language.
>
>
> [1] Lachaux, Marie-Anne, et al. "Unsupervised translation of programming languages." *arXiv preprint arXiv:2006.03511* (2020).
>
> [2] Kusner, Matt, et al. "From word embeddings to document distances." *International conference on machine learning*. PMLR, 2015.
>
> ------
> **Sample 1** (Scala, Go)
> ```
> import scala.io.StdIn
>
> object Main extends App {
>
>   val n = StdIn.readLine().trim.toInt
>   class Point(val x: Double, val y: Double){
>     override def toString: String = {
>       f"$x%.8f $y%.8f"
>     }
>   }
>
>   final val start = new Point(0, 0)
>   final val end   = new Point(100, 0)
>   final val th = scala.math.Pi * 60.0 / 180.0
>
>   def calcuPoint(p1: Point, p2: Point, a: Double, b: Double) = new Point((a*p1.x+b*p2.x)/3.0, (a*p1.y+b*p2.y)/3.0)
>
>
>   def koch(d: Double, p1: Point, p2: Point): Unit = {
>     if(d != 0){
>       val s = calcuPoint(p1, p2, 2.0, 1.0)
>       val t = calcuPoint(p1, p2, 1.0, 2.0)
>       val u = new Point((t.x - s.x)*math.cos(th) - (t.y - s.y)*math.sin(th) + s.x,
>                         (t.x - s.x)*math.sin(th) + (t.y - s.y)*math.cos(th) + s.y)
>       koch(d-1, p1, s)
>       // print s
>       println(s)
>
>       koch(d-1, s, u)
>       // print u
>       println(u)
>
>       koch(d-1, u, t)
>       // print t
>       println(t)
>
>       koch(d-1, t, p2)
>
>     }
>   }
>
>   println(start)
>   koch(n, start, end)
>   println(end)
> }
> ```
>
>
> ```
> package main
>
> import (
> 	"bufio"
> 	"fmt"
> 	"math"
> 	"os"
> 	"strconv"
> )
>
> type Point struct {
> 	x float64
> 	y float64
> }
>
> // d: deepness of recursion
> // p1, p2: points of edges
> func koch(d int, p1, p2 Point) {
> 	if d == 0 {
> 		return
> 	}
>
> 	var s, t Point // Points equally divided into 3
> 	var u Point    // Vertex of equilateral triangle
>
> 	// calc point of s, u, t by using values of p1 and p2
> 	th := math.Pi * 60.0 / 180.0
> 	s.x = (2.0*p1.x + 1.0*p2.x) / 3.0
> 	s.y = (2.0*p1.y + 1.0*p2.y) / 3.0
> 	t.x = (1.0*p1.x + 2.0*p2.x) / 3.0
> 	t.y = (1.0*p1.y + 2.0*p2.y) / 3.0
> 	// rotation matrix
> 	u.x = (t.x-s.x)*math.Cos(th) - (t.y-s.y)*math.Sin(th) + s.x
> 	u.y = (t.x-s.x)*math.Sin(th) + (t.y-s.y)*math.Cos(th) + s.y
>
> 	koch(d-1, p1, s)
> 	fmt.Println(s.x, s.y)
> 	koch(d-1, s, u)
> 	fmt.Println(u.x, u.y)
> 	koch(d-1, u, t)
> 	fmt.Println(t.x, t.y)
> 	koch(d-1, t, p2)
> }
>
> func main() {
> 	sc := bufio.NewScanner(os.Stdin)
> 	var p1 = Point{
> 		x: 0,
> 		y: 0,
> 	}
> 	var p2= Point{
> 		x: 100,
> 		y: 0,
> 	}
> 	sc.Scan()
> 	d, _ := strconv.Atoi(sc.Text())
>
> 	fmt.Println(p1.x, p1.y)
> 	koch(d, p1, p2)
> 	fmt.Println(p2.x, p2.y)
> }
> ```
>
> -------
> **Sample 2 (COBOL, Python)**
> ```
> IDENTIFICATION DIVISION.
> program-id. kyopuro.
>
> data division.
> working-storage section.
> 01 in-str pic x(22).
> 01 n pic 9(10).
> 01 m pic 9(10).
> 01 res pic 9(19).
> 01 t pic 9(5).
> 01 view-res pic z(18)9.
>
> procedure division.
> main.
>     accept in-str.
>     unstring in-str delimited by all SPACE
>     into n m.
>
>     if n = 1 and m = 1
>     then
>         move 1 to view-res
>     else if n = 1 or m = 1
>     then
>         compute view-res = function max(n, m) - 2
>     else
>         compute view-res = (n - 2) * (m - 2)
>     end-if
>     end-if.
>
>     display view-res.
>     stop run.
> ```
>
> ```
> # -*- coding: utf-8 -*-
>
> n, m = map(int, input().split())
>
> if n == 1:
>     if m == 1:
>         ans = 1
>     else:
>         ans = m-2
> else:
>     if m == 1:
>         ans = n-2
>     else:
>         ans = (n-2)*(m-2)
>
> print(ans)
> ```

---

### Decision · Program_Chairs · 2022-01-20

**Decision:**

Reject

**Comment:**

Motivated by addressing the problem of lacking parallel training data for supervised code translation, this paper proposed to construct noisy parallel source code datasets using a document similarity based approach, and empirically evaluated its effectiveness for code translation tasks.

The paper is in general well-written, easy to follow, and the method is simple and empirical results look positive. Some major concern by reviewers is that while the proposed method is simple and may be easy to use, the overall technical novelty/contribution is limited, e.g., there generally lacks of more thorough discussions on how to deal with the critical noise issue in a more robust or sophisticated way. In addition, there were also other concerns about the experimental issues, such as datasets, metrics, ablation analysis, usability, etc.

Overall, the paper presents some preliminary positive results for an interesting research problem, but the overall technical novelty and contributions are incremental and the paper is not strong enough for the acceptance by this conference. Nonetheless, this work could be potentially valuable for the niche area of code translation research, and authors are encouraged to continue to improve this research with more thorough investigation for a future venue.